# Measurement and Calibration of Regulatory Credit Risk Asset Correlations

Anton van Dyk [1,2] and Gary van Vuuren [3],*

1   Department of Mathematics and Applied Mathematics, University of Pretoria, Pretoria 0028, South Africa; anton.vandyk@riskworx.com
2   RiskWorx, Johannesburg 2031, South Africa
3   Centre for Business Mathematics and Informatics, Potchefstroom Campus, North-West University, Potchefstroom 2520, South Africa
*   Correspondence: vvgary@hotmail.com

**Abstract:** Vasicek's asymptotic single risk factor (ASRF) model is employed by the Basel Committee on Banking Supervision (BCBS) in its internal ratings-based (IRB) approach for estimating credit losses and regulatory credit risk capital. This methodology requires estimates of asset correlations; these are prescribed by the BCBS. Practitioners are interested to know market-implied asset correlations since these influence economic capital and lending behavior. These may be backed out from ASRF loan loss distributions using ex post loan losses. Prescribed asset correlations have been neither updated nor recalibrated since their introduction in 2008 with the implementation of the Basel II accord. The market milieu has undergone significant alterations and adaptations since then; it is unlikely that these remain relevant. Loan loss data from a developed (US) and developing (South Africa) economy spanning at least two business cycles for each region were used to explore the relevance of the BCBS calibration. Results obtained from three alternative methodologies are compared with prescribed BCBS values, and the latter were found to be countercyclical to empirical loan loss experience, resulting in less punitive credit risk capital requirements than required in market crises and more punitive requirements than required in calm conditions.

**Keywords:** asset correlation; loan losses; asymptotic single risk factor model

## 1. Introduction

Credit portfolios comprise an assortment of financial debt instruments—such as bonds and loans—held by banks and financial institutions; credit risk arises from the possibility of loss when obligors fail to meet repayment obligations. Loss data (such as the severity of losses given default and exposure at the time of default) are used in credit models developed from pioneering work by Merton (1974) and Black and Cox (1976), and adapted and updated at various times by, e.g., Alfonsi and Lelong (2012) and Cohen and Costanzino (2017). Credit loss distributions arising from empirical losses (the characteristics of which play a vital role in predicting credit-risky portfolio behavior) have been described by various authors (see, for example, Crouhy et al. 2000 and sources therein), but it was Vasicek (1987, 2002, 2015) who laid the foundations for the rules governing the calculation of regulatory credit risk capital (BCBS 2005). Banks are required to mitigate credit risk by allocating sufficient capital to absorb credit losses when defaults or downgrades occur using the Vasicek framework.

The prevailing approach to model default correlation combines default probabilities with asset correlations. This methodology involves linking defaults of two borrowers to their asset values, in which insufficient assets lead to simultaneous defaults of both borrowers. This idea, introduced by Vasicek (1987) gained traction due to its reliance on continuously available market data, bypassing limitations of historical default information. It underpins various credit risk models, including the BCBS credit risk capital charge's

ASRF model. For this BCBS-mandated approach, banks may measure and use as input into the prescribed ASRF approach some of the required input parameters such as probabilities of default and losses given default (after satisfying relevant supervisory criteria). Other parameters such as asset correlation, $\rho$, are fixed by the BCBS regardless of the approach adopted. The asset $\rho$ measures the degree of co-movement between obligors' loan health, while $\sqrt{\rho}$ measures the degree of the obligor's co-movement with the single systematic risk factor (BCBS 2005; Zhang et al. 2008) to which all borrowers are linked. This parameter ($\rho$) is difficult to measure because of a lack of required sector data and often incomparable obligor characteristics. As a result, banks may not determine this value themselves but are instead forced to rely on compulsory calibration and fixing by the BCBS (BCBS 2005, 2023). The $\rho$ values stipulated by the BCBS were determined prior to the 2008 credit crisis and have, to date (August 2023), never been altered, despite the severity of the credit crisis (which led to severe widespread defaults) or the impact of the COVID-19 pandemic of 2020–2021 which introduced another critical shock to the global economy.

Market-implied asset $\rho$s may, however, be estimated using relevant loan loss data: there are several approaches to accomplish this. How these market-implied $\rho$s have altered since their institution in 2008 provides interesting insights into credit risky behavior and inform whether the Basel II values remain sensible or even economically feasible.

The remainder of this article proceeds as follows: Section 2 sets out a combination of a review of historical work pertaining to this research and the theoretical developments of the relevant mathematics. This covers the Vasicek asymptotic single risk factor model development, the use of the beta distribution in credit risk loss assessment, and the techniques required to reverse-engineer asset correlations from known loan losses. Section 3 discusses the data used, provides the rationale for their use in this exercise, and extends the theoretical discussion from the preceding section. Section 4 presents the results and discusses possible consequences. Section 5 concludes the paper.

## 2. Theoretical Development and Literature Review

Credit risk is the potential for financial loss caused by a borrower's inability to make payments on debt obligations. Portfolios of many loans—aggregated by loan type—behave in predictable ways, which has led to the development of credit risk models to evaluate loan portfolio losses. The earliest credit risk models were based on financial ratios, such as the debt-to-equity (or asset) ratio and the interest coverage ratio, to predict default likelihood. These models were, however, limited in their ability to predict default accurately, leading to the development of more sophisticated credit risk models in the 1980s and 1990s, with a shift toward the use of statistical techniques, such as logistic regression and survival analysis (Crouhy et al. (2000) provided a comprehensive review of credit risk model evolution).

A widely used contemporary credit risk model described by Merton (1974) employs the Black–Scholes option pricing formula to estimate the value of a firm's debt and equity, and then calculates the probability of default as a function of the difference between the value of the assets and the value of the liabilities. The Merton model is widely used in corporate finance, but its assumption of normally distributed returns and constant volatility has been criticized as unrealistic (see e.g., Majumder 2006).

Also widely used is the KMV model which uses a firm's historical stock price data to estimate the probability of default. The model assumes that the firm's stock price reflects its creditworthiness, and that a sharp decline in the stock price is an indicator of an increased risk of default. The KMV model has been criticized for its reliance on historical stock prices, which may not accurately reflect a firm's current creditworthiness, and its reliance on the assumption that asset values are normally distributed (Zhan et al. 2013).

Recent improvements in credit risk modeling have been instituted using machine learning techniques, such as artificial neural networks and support vector machines. These models are more accurate in predicting default than traditional statistical models, but their widespread adoption is still in its relative infancy; thus, further research is needed to fully understand their potential benefits and limitations (Khandani et al. 2010; Shi et al. 2022).

Vasicek's (1987) work on single and multifactor credit risk models dominate the banking credit risk landscape. The ASRF model (Vasicek 2002) is a structural mathematical model describing the mechanics of the default process, which has been adopted by the BCBS in the regulatory credit risk framework. Unlike reduced-form models (which focus on the statistical properties of default), the Vasicek (2002) ASRF model assumes that the default process is mean-reverting, i.e., that PDs evolve over time depending on the spread between the risk-free rate and borrower's credit spread (Kupiec 2007), and that credit portfolios become invariant to new obligors as the obligor number becomes large. Widely used in the financial industry because of its computational efficiency, it provides a simple, tractable credit risk assessment framework. Inevitably, it also has some limitations such as the assumption of the market index as a proxy for the economy and portfolio invariance for large obligor numbers (Gordy 2003; Cowan and Cowan 2004; García-Céspedes and Moreno 2017).

Zhou (2001) developed an elementary theoretical framework describing default correlations using the concept of first-passage times. A company is considered to default once its value initially crosses a predefined default threshold. Estimating the default correlation between two companies involves computing the likelihood of a two-dimensional random process crossing a threshold. Zhou (2001) demonstrated the way in which asset-return correlations, default correlations, and time horizons are interconnected but these assertions have been largely discredited. For example, Li and Krehbiel (2016) asserted an inconsistency between the stochastic assumptions of Merton's firm-specific default probability model with the bivariate first passage time model of default correlation and derived a closed-form equation to determine default correlations. Accornero et al. (2018) demonstrated how the ASRF model could be disadvantageous because of the reliance on a single risk factor. The asset correlation errors resulting from simplified single risk factor models lead to considerable default correlation errors. Mwamba et al. (2019) used similar approaches to Stoffberg and van Vuuren (2015) and closely corroborated their results using South African loan loss data. They suggested that, because of the discrepancies between empirical asset correlations and those prescribed by the BCBS, systematic contagion could result since all banks in a jurisdiction would be similarly undercapitalized. Mwamba et al. (2019) urged central banks to consider the findings and adapt or revise their asset correlation values.

Although it is widely accepted that observed default rates or even equity returns may be used to calibrate a single factor Gaussian copula model (as is the case for the ASRF), the formulation is still likely to understate tail risk (Dias 2020). Using a Bayesian approach in which asset correlations are modeled using an inverse Wishart prior and equity correlations to obtain the posterior distribution, Dias (2020) found that probabilistic forecasts of defaults were produced with better out-of-sample performance than the standard ASRF model. Cho and Lee (2022) used a time-varying credit risk model to extract empirical asset correlations from loan loss data (the identical dataset to that used in our work). The model outperformed the regulatory model for US credit portfolios with strong empirical evidence of cyclical and asymmetric asset correlation. The authors argued that Basel's mandatory criteria for determining asset correlation was insufficient during economic downturns.

The Basel II Framework was instituted in most compliant countries in January 2008 (BCBS 2004). The aim of the framework was to provide stability in the risk management industry, including the credit risk sector, strengthen regulatory capital requirements, and support good banking practices (Stephanou and Mendoza 2005). The framework requires, inter alia, two main approaches that banks may use to determine regulatory credit risk capital: the standardized and the internal ratings-based (IRB) approaches (BCBS 2006).

The standardized approach to credit risk presents a method of determining the minimum capital requirement for credit risk according to a standardized set of rules and inputs that are used to determine the level of risk associated with different types of exposures. The minimum capital requirement involves calculating the product of each exposure and a prescribed risk weight (based on credit quality of the exposure). The approach includes guidelines for determining the credit quality of exposures, as well as measuring and moni-

toring credit risk and promotes a more consistent and transparent approach to credit risk management across the banking industry (BCBS 2006).

The IRB approach uses Vasicek's (2002) ASRF model and regulates banks to categorize loans according to specific criteria. Each category comprises unique characteristics, including risk components and minimum capital requirements (BCBS 2006). These risk components are the PD, the exposure at default (EAD), the loss given default (LGD), loan maturity, *M*, and asset correlations $\rho$. The resultant calculations determine the minimum regulatory credit risk capital requirements (Gordy 2003).

The expected loss of a credit portfolio is the total expected loss a credit portfolio holder can expect to experience on their portfolio over a chosen time horizon (Chatterjee 2015). The expected loss (EL) is calculated as the aggregate sum of the expected loss for each obligor in a portfolio of *N* obligors:

$$EL = \sum_{i=1}^{N} PD_i \cdot LGD_i \cdot EAD_i$$

Under the IRB approach, credit portfolio risk is the unexpected loss ($UL_p$), determined using the standard deviation of credit losses (Chatterjee 2015):

$$UL_p = \sum_{i=1}^{N} \sigma_i \cdot \rho_{ip}$$

where $\sigma_i$ is the standard deviation of credit losses for obligor *i*, and $\rho_{ip}$ denotes the $\rho$ between obligor *i* and overall portfolio *p* credit losses.

The arrangement of these credit losses (EL and UL) in the Vasicek credit loss distribution is shown in Figure 1.

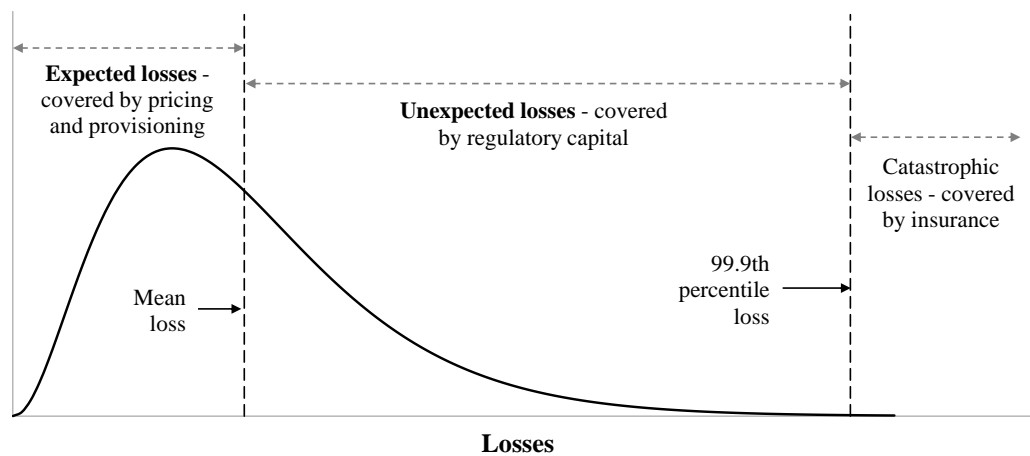

**Figure 1.** The credit loss distribution. Source: authors' representation.

Under the IRB's ASRF approach, using the Vasicek formulation, the minimum capital requirement (*K*) for corporate, sovereign, bank, and high-volatility commercial real-estate exposures is given by

$$K = EAD \cdot LGD \cdot \left( \Phi \left[ \frac{\Phi^{-1}(PD) + \sqrt{\rho} \cdot \Phi^{-1}(0.999)}{\sqrt{1-\rho}} \right] - PD \right) \cdot \underbrace{\frac{1 + (M - 2.5) \cdot b}{1 - 1.5 \cdot b}}_{\text{Maturity adjustment}} \quad (1)$$

where *M* is the effective (remaining) maturity of the obligation in years (floored at one year and capped at five), *b* is a scaling coefficient dependent only on PD, and a systematic factor of 0.999 implies a 99.9% confidence level or the credit risk capital required to cover the

annual unexpected losses arising from a 99.9th worst-case systematic factor scenario as required by the BCBS (2005).

Banks may use their own internal models to determine $PD$, $LGD$, $EAD$, and $M$, but independent estimation of asset $\rho$ is not permitted. Set by the BCBS (2005), these are either fixed or dependent on PDs and are set out in Table 1. Other loans, such as mortgages, as well as revolving and other retail, do not include the maturity adjustment used in (1). Developing a method to determine the 'true' asset $\rho$—as established by market participants and embedded in credit portfolios—is crucial in credit risk measurement.

**Table 1.** Asset $\rho$ for various exposures under the IRB ARSF approach. Source: BCBS (2005, 2023).

| Loan Type | Asset $\rho$ |
|---|---|
| Residential mortgage | 15% |
| Qualifying revolving retail | 4% |
| Other retail | $0.03 \cdot \left( \frac{1-e^{-35 \cdot PD}}{1-e^{-35}} \right) + 0.16 \cdot \left( 1 - \left[ \frac{1-e^{-35 \cdot PD}}{1-e^{-35}} \right] \right)$ |
| Corporates, sovereigns, and banks | $0.12 \cdot \left( \frac{1-e^{-50 \cdot PD}}{1-e^{-50}} \right) + 0.24 \cdot \left( 1 - \left[ \frac{1-e^{-50 \cdot PD}}{1-e^{-50}} \right] \right)$ |
| High-volatility commercial real estate | $0.12 \cdot \left( \frac{1-e^{-50 \cdot PD}}{1-e^{-50}} \right) + 0.30 \cdot \left( 1 - \left[ \frac{1-e^{-50 \cdot PD}}{1-e^{-50}} \right] \right)$ |

In the ASRF model, the singular systematic risk factor takes on the role of an indicator for the global economic state. This pivotal factor is gauged by the asset correlation, $\rho$, signifying the extent of an obligor's susceptibility to systematic risk. This correlation delineates the interrelationship between the asset value of one borrower and that of another, capturing how borrowers' asset values hinge on the overall economic health. All borrowers are inherently intertwined through this pivotal risk factor. These asset correlations form the bedrock for establishing the BCBS risk weight functions, inherently contingent on asset classes due to varying dependencies of different borrowers and asset categories on the broader economic panorama. Dissimilar asset correlations emerge from analyzing divergent loss experiences among portfolios that share identical anticipated losses. When correlation prevails among individual exposures within a portfolio and with the overarching systematic risk factor of the ASRF model, the result is a heightened variance in loss rates. This configuration mirrors a portfolio marked by elevated interactions between borrowers, where defaults are tightly knit to the prevailing economic climate (BCBS 2005, 2023).

In (1), $\rho$ measures the degree of co-movement between obligors' loan health, while $\sqrt{\rho}$ measures the degree of the obligor's exposure to the systematic risk factor. Because this parameter is difficult to estimate and, if unrestricted, could provide banks with too much model flexibility leading to widely disparate risk weightings and associate credit risk capital, the IRB approach specifies and prescribes asset $\rho$ coefficients, given in Table 1 (BCBS 2023). Note that these are unchanged since the introduction of the prescribed asset correlations in 2005 (BCBS 2005). These are provided for various loan types and are either fixed or vary with PD. Those that vary with PD do so monotonically (Figure 2), decreasing as PD increases (Gordy 2003; Lopez 2004). In Table 1, the value which precedes the first bracket is the lower bound $\rho$, and the value which precedes the second bracket is the upper bound.

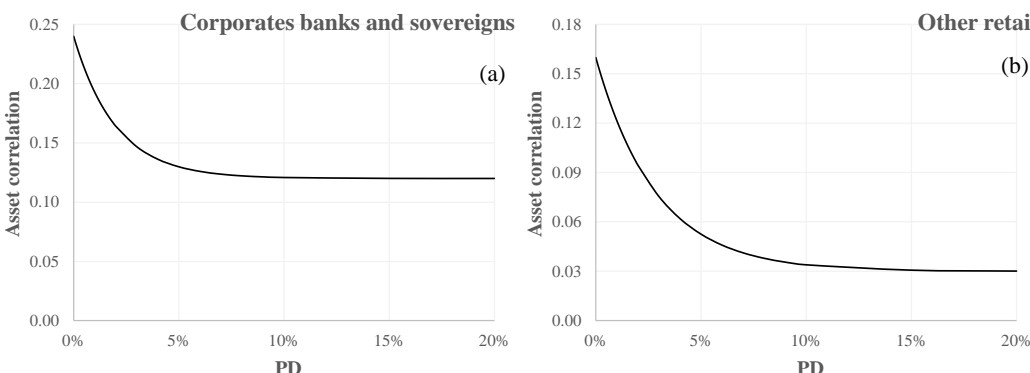

**Figure 2.** Asset $\rho$ dependence on PD for (**a**) corporate, bank, and sovereign loans, and (**b**) other retail.

The rationale for the asset $\rho$ profiles in Figure 2 arises from the empirical observations that different loans or obligors with low PDs have higher asset $\rho$s and vice versa. Dev (2006) explained this feature intuitively, invoking the sensible reasoning that loans with lower PDs are generally larger and more dependent on the economy or market (i.e., the systemic factor) as a whole. Loans with higher PDs are more influenced by idiosyncratic factors whilst those loans with lower PDs are more impacted by systematic factors. This heuristic explanation assumes different loans in an economy; at any specific time, different asset $\rho$s are assigned to different loans, dependent only on their respective PDs. The logic of this framework does not, however, extend to large portfolios of heterogeneous loans. Consider the evolution of such a large loan's average portfolio PD evolving over time. Changes in average portfolio PD are caused by specific market circumstances and shifts in the economic milieu. A worsening economic environment increases the average portfolio PD because incipient loans' asset values—being correlated with the systemic factor—deteriorate together (albeit to different extents).

As an example, consider a bank's portfolio of many debt securities prior to the credit crisis of 2008/2009. In the benign economic environment which preceded the crisis, default rates of such a portfolio were low. As market conditions deteriorated, default rates increased considerably as all obligors—correlated with the systemic factor—suffered increased impairments (thereby increasing average portfolio PD). It is not counterintuitive to envision a scenario in which incumbent asset $\rho$s lead to a higher portfolio PD which increases asset $\rho$s as panic sets in, binding loan quality even more tightly to the market environment and so on, in a malign feedback loop.

Tasche (2008) posited that high levels of asset $\rho$s observed during the crisis were a major contributor to the high default rates observed in loan portfolios and that those banks which had relied on diversification as a risk management tool were particularly vulnerable to the breakdown in diversification that occurred during the crisis. Acharya and Richardson (2009) argued that the high levels of asset $\rho$s observed during the crisis eroded diversification as different types of assets became highly correlated with the market and each other, leading to higher loan portfolio default rates as the risk of individual loans became more difficult to manage.

Tarashev and Zhu (2018) explored how specification and calibration errors can impact portfolio credit risk measurement accuracy, specifically in the ASRF model. They concluded that these errors can significantly affect the model's performance and that proper calibration was crucial for accurate risk measurement. Lee et al. (2011) assessed the asymmetric behavior and procyclical impact of asset correlations. In an investigation of the relationship between asset correlations and economic cycles, correlations were found to behave differently during economic expansions and contractions. High correlations during expansions are associated with a procyclical impact on the economy, while low correlations during contractions lead to a countercyclical impact.

Half a decade before the crisis, Duchemin et al. (2003) found the BCBS-prescribed asset $\rho$s to be conservative and suggested that the volatility of the PD parameter should

be included in estimating the empirical asset $\rho$. This conservatism in asset $\rho$ values was echoed by Hartigan (2003) in a stinging letter to the BCBS.

Using empirical data, Zhang et al. (2008) found that, despite reasonable agreement between their default implied asset $\rho$s and those specified in the Basel II Accord for large corporate borrowers, other default-implied asset $\rho$ values were considerably higher than previous studies even after applying the small corporates adjustment. Measuring asset $\rho$ ex ante led to the improvement of realized default $\rho$s and portfolio credit risk, statistically and economically. The authors asserted that important practical, economic, and regulatory implications stemmed from the discrepancies between Basel-specified asset $\rho$s and market-implied results.

Botha and van Vuuren (2010) concluded that the BCBS estimates of asset $\rho$ outlined in Table 1 were conservative compared with market-implied estimates, measured by reverse engineering (1) and using US credit loss data. Market-implied data suggested that asset $\rho$s were much lower in the US than the values provided by the BCBS. The authors concluded that this deviation introduced a level of conservatism in the asset $\rho$ parameter and allowed for more prudent credit risk measurements but should be monitored regularly in case the mandated and market-implied values continue to drift apart.

## 3. Materials and Methods

### 3.1. Materials

For the US, national, quarterly, non-seasonally adjusted charge-off rates (all loan types from all insured US chartered commercial banks were used as historical data to estimate the market-implied asset $\rho$ (FRED Economic Data 2023). Charge-off rates are loan and lease values removed from bank books and charged against loss reserves; hence, these serve as a convenient proxy for sovereign-wide annualized losses net of recoveries and measured as a percentage of average loans. Delinquent loans are those past due more than 90 days (including those with interest accrual and nonaccrual status) measured as a percentage of end-of-period loans. Loans for residential real estate encompass those that are secured by properties with one to four families, which can include home equity lines of credit, while loans for commercial real estate consist of construction and land development loans, loans secured by multi-family residences, and loans secured by non-farm, non-residential real estate (FRED Economic Data 2023).

Charge-off rate data were assembled from the following:

- Q1-85 to Q4-22 for qualifying revolving retail (credit card), other retail and corporate/sovereign and bank exposures, and
- Q1-91 to Q4-22 for residential mortgage and high volatility commercial real estate exposures.

South African charge-off rate data were collected from Marsh global loss data (Marsh 2023) over the period spanning January 1993 to July 2022. These data represent country-wide charge off rates (all loan types). Because principal bank loan losses arise from commercial and industrial loan type in South Africa, the standards applied to this loan type were instituted in the analysis which follows.

We are aware that the span of our data predates the Basel regulatory rules for determining the requisite credit risk capital (which were only required from 2008 with the institution of Basel II (BCBS 2006)). This work, then, investigates not only the requisite credit risk capital since the initiation of Basel II, but also that which *would have been required* had the Basel II rules been in place in the 1980s.

### 3.2. Methods

#### 3.2.1. The Vasicek ASRF Model

The original framework for credit loss distributions was given by Vasicek (2002). The derivations and results formulated by Vasicek (2002) laid the foundation for credit risk research and provided much-needed risk analysis methods for banks. Vasicek (2002) made several assumptions regarding credit loss portfolios and their behavior. The first assump-

tion is that a loan defaults if the respective borrower's assets fall below the obligation payment of the borrower. This assumption gives rise to the first derivation, the probability of default on the *i*-th loan can be represented by $P(A_i < B_i)$, where $A_i$ is the asset value of obligor *i* and $B_i$ is the loan amount of obligor *i*.

The asset value of obligor *i* evolves in time according to the stochastic process:

$$dA_i = \mu_i A_i dt + \sigma_i A_i dX_i$$

where $X_i$ satisfies standard one-dimensional Brownian motion, $W_t$, $\mu_i$ is the mean, and $\sigma_i$ is the standard deviation of obligor *i*'s asset returns. Considering this process, the discrete asset value of obligor *i* is

$$\log(A_i(T)) = \log(A_i) + \left(\mu_i - \frac{\sigma_i^2}{2}\right)T + \sigma_i X_i \sqrt{T} \tag{2}$$

The *PD* of obligor *i* is $PD_i = P[A_i(T) < B_i] = P(X_i < c_i) = N(c_i)$, where

$$c_i = \frac{\log(B_i) - \log(A_i) - \left(\mu_i T - \frac{\sigma_i^2}{2}\right)T}{\sigma_i \sqrt{T}}$$

and $N$ is the cumulative normal distribution function. $X_i$ in (2) is described by

$$X_i = Y\sqrt{\rho} + Z_i\sqrt{1-\rho}$$

where $Y$, $Z_i$ are mutually independent standard normal variables with $Y$ representing the systematic risk factor such as an economic index, and $Z_i$ representing a borrowing company's specific risk factor (any risk that affects the company uniquely such as client base behavior, operational risk and revenue); $\rho_i$ is the asset $\rho$ between the *i*-th asset return and the systematic risk factor.

Vasicek (2002) considered a homogeneous portfolio comprising *n* number of loans with equal loan exposures and PDs, the same maturity, $T$ and the correlation between any two obligor asset values $= \rho$. Defining $l_i$ as an indicator function representing default of obligor *i* such that $l_i = 0$ if obligor *i* defaults and $l_i = 1$ if obligor *i* does not default, the portfolio percentage loss, $L$, is

$$L = \frac{1}{N}\sum_{i=1}^{N} l_i$$

Since the events of default are not independent, the central limit theorem does not hold, and the loss distribution does not converge to a limit form. If the systematic risk factor, $Y$, is known, the conditional probability of loss on a single loan is

$$
\begin{aligned}
PD(Y) &= P(D_i = 1 | Y = y) = P(A_i(T) < B_i | Y = y) \\
&= P(X_i < c_i | Y = y) = P(\sqrt{\rho}Y + \sqrt{1-\rho}Z_i < c_i) \\
&= P\left(Z_i < \frac{c_i - \sqrt{\rho}Y}{\sqrt{1-\rho}} | Y = y\right) \\
&= N\left(\frac{N^{-1}(PD) - \sqrt{\rho}Y}{\sqrt{1-\rho}}\right)
\end{aligned}
\tag{3}
$$

$PD(Y)$ is the loan default probability given the scenario $Y$. The variables $l_i$, conditional on $Y$, are independent equally distributed variables with finite variance. Thus, the portfolio percentage loss $L$, given $Y$, converges by the law of large numbers to its expectation $PD(Y)$ as $N \to \infty$. Therefore,

$$P(L \leq x) = P(PD(Y) \leq x) = P\left(Y \geq PD^{-1}(x)\right) = N\left(-PD^{-1}(x)\right)$$

Allowing the number of loans on a given portfolio to approach infinity, and through the law of large numbers, Vasicek (2002) derived the main result of his proposed distribution. The cumulative distribution function of loan losses on a very large portfolio:

$$P(L \leq x) = N\left(\frac{\sqrt{1-\rho} \cdot N^{-1}(x) - N^{-1}(PD)}{\sqrt{\rho}}\right) \qquad (4)$$

Vasicek (2002) then developed the loss distribution function of a very large portfolio of loans:

$$f(x; PD; \rho) = \sqrt{\frac{1-\rho}{\rho}} \cdot \exp\left[\frac{1}{2}\left(N^{-1}(x)\right)^2 - \frac{1}{2\rho}\left(N^{-1}(x)\sqrt{1-\rho} - N^{-1}(PD)\right)^2\right] \qquad (5)$$

### 3.2.2. Vasicek ASRF Portfolio Loss Distribution: Mode

This density function of the Vasicek ASRF portfolio loss distribution (5) is unimodal with mode

$$L_{mode} = N\left[\frac{\sqrt{1-\rho}}{1-2\rho} \cdot N^{-1}(PD)\right] \qquad (6)$$

(6) may be manipulated (Botha and van Vuuren 2010; Stoffberg and van Vuuren 2015) to determine an estimator for the asset $\rho$ coefficient:

$$\frac{N^{-1}(L_{mode})}{N^{-1}(PD)} = \frac{\sqrt{1-\hat{\rho}}}{1-2\hat{\rho}}\left(\frac{N^{-1}(L_{mode})}{N^{-1}(PD)}\right)^2 = \frac{1-\hat{\rho}}{(1-2\hat{\rho})^2}$$

Thus, we can define

$$\psi = \left(\frac{N^{-1}(L_{mode})}{N^{-1}(PD)}\right)^2$$

which gives $\psi(1-2\hat{\rho})^2 = 1 - \hat{\rho}$, or, in quadratic form,

$$4\psi\hat{\rho}^2 + (1-4\psi)\hat{\rho} + (\psi-1) = 0 \qquad (7)$$

Solving for $\rho$ in (7) gives

$$\hat{\rho} = \frac{(4\psi-1) \pm \sqrt{8\psi+1}}{8\psi} \qquad (8)$$

Due to the quadratic form of (7), there are two possible solutions for the asset $\rho$ estimator, but only the negative solution produces a mathematically tractable result. Extracting the asset $\rho$ involves assembling gross credit loss data as a percentage of the total loan value and determining the mode of the resulting loss distribution by measuring the average gross loss as a proportion of the total loan value. Knowing $L_{mode}$, (8) may be used to extract the market implied $\rho$ embedded in known ex post loan losses.

### 3.2.3. Vasicek ASRF Portfolio Loss Distribution: Variance

The mean of the Vasicek ASRF portfolio loss distribution (5) is $EL = PD$ and the variance, $s^2$, is $s^2 = E(x^2) - [E(x)]^2$ or

$$s^2 = \int_{-\infty}^{\infty} x^2 \cdot f(x) dx - PD^2$$

$$s^2 = \int_{-\infty}^{\infty} x^2\left(\frac{\hat{\rho}}{1-\hat{\rho}}\right)\exp\left(-\frac{1}{2\hat{\rho}} \cdot \left(N^{-1}(x) \cdot \sqrt{1-\hat{\rho}} - N^{-1}(PD)\right)^2 + \frac{1}{2}\left(N^{-1}(x)\right)^2\right) dx - PD^2 \qquad (9)$$

The method to extract $\hat{\rho}$ is, thus, as follows (again using gross credit loss data as a percentage of the total loan value): determine the empirical variance, $s$, of relevant loan losses, and reverse-extract $\hat{\rho}$ using (9) since this is then the only unknown variable, extracted using a numerical integration approach.

### 3.2.4. $\beta$ Distribution Fitting

The method proposed by Hansen et al. (2008) fits a $\beta$ distribution to observed annualized loss rates, which provides an estimate of credit losses at a 99.9% confidence level (chosen to be that required by regulatory requirements).

A $\beta$ distribution is completely characterised by two parameters, $\alpha$ and $\beta$, which are easily obtained from the mean ($\mu$ = EL) and standard deviation ($\sigma$) of the loan losses. These quantities are linked by

$$\alpha = \mu \cdot \left( \frac{\mu \cdot (1 - \mu)}{\sigma^2} - 1 \right) \quad \beta = (1 - \mu) \cdot \left( \frac{\mu \cdot (1 - \mu)}{\sigma^2} - 1 \right)$$

The probability density function for a $\beta$ distribution is

$$f(x, \alpha, \beta) = \frac{\Gamma(\alpha + \beta)}{\Gamma(\alpha) \cdot \Gamma(\beta)} \cdot \left[ (1 - t)^{\beta - 1} \cdot t^{\alpha - 1} \right] \tag{10}$$

where $\forall \, \alpha, \, \beta > 0$ and $1 \geq x \geq 0$. The cumulative density function for the $\beta$ distribution, $B(\alpha, \beta, x)$, may be used to calculate losses at any given percentile value, $x$, say 99.9% as per Basel regulatory requirements. Thus, equating (3) with the cumulative density function for the $\beta$ distribution gives

$$B(\alpha, \beta, x) = N \left( \frac{N^{-1}(PD) + \sqrt{\rho} \cdot N^{-1}(99.9\%)}{\sqrt{1 - \rho}} \right)$$

$$N^{-1}[B(\alpha, \beta, x)] = \frac{N^{-1}(PD) + \sqrt{\rho} \cdot N^{-1}(99.9\%)}{\sqrt{1 - \rho}} \tag{11}$$

Let $v = N^{-1}[B(\alpha, \beta, x)]$, $p = N^{-1}(PD)$, and $q = N^{-1}(99.9\%)$.
Thus, $v \cdot \sqrt{1 - \hat{\rho}} = p + q \cdot \sqrt{\hat{\rho}}$.
After some tedious algebra, (11) may be rewritten as

$$\hat{\rho} = \frac{(v^2 + p^2) \cdot (v^2 + q^2) - 2vp \cdot \left( vp + q\sqrt{v^2 + q^2 - p^2} \right)}{(v^2 + q^2)^2} \tag{12}$$

where (12) measures the market implied asset $\rho$ provided by Hansen et al. (2008).

There are, thus, three approaches to measuring the market-implied asset $\rho$ as illustrated in (8), (9), and (12). These form the basis of the comparisons with the asset $\rho$s as given by the BCBS, i.e., Table 1.

## 4. Results

The charge-off rates for all US bank exposures are shown in Figure 3. Charge-off rates (hence, default rates) peaked at various times over the 30-year data span in both countries.

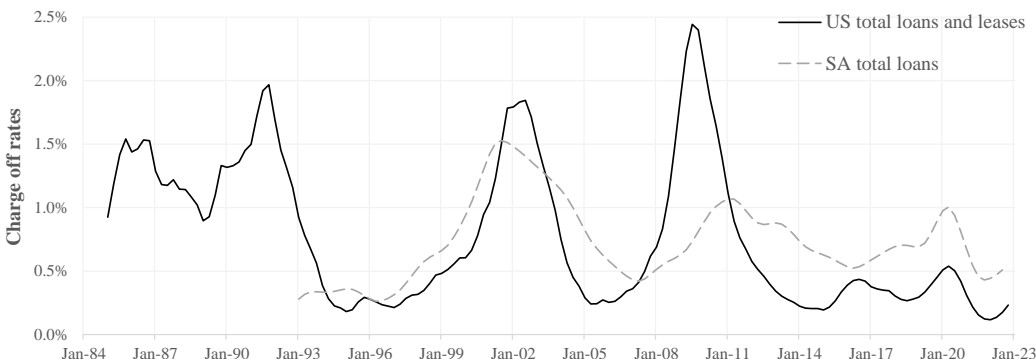

**Figure 3.** Charge-off rates on all loan exposures from commercial banks in the US and South Africa. Source: FRED Economic Data (2023) and Marsh (2023).

The surge of US defaults that occurred in

- 1991/1992 arose due to the 1990/1991 recession exacerbated by the 1990 oil price shock and high levels of US consumer and corporate debt,
- 2002 were caused by the bursting of the dot-com bubble in the early 2000s, the 9/11 terrorist attacks in 2001, and the accounting scandals at Enron and WorldCom in 2002, which exposed widespread financial fraud and accounting irregularities, and
- 2009/2010 developed because of the global financial crisis which began in 2008, triggered by the collapse of the US housing market, subprime mortgage defaults, and the widespread use of risky financial derivatives.

The increase in South African defaults that occurred in

- 2001 was due in part to the global economic slowdown, but also because of spillover effects caused by the crisis in neighboring Zimbabwe which had a significant impact on South Africa's political and economic stability,
- 2011 occurred because of the European debt crisis, which led to a significant drop in demand for South African exports to Europe and a decrease in foreign investment, as well as the mismanagement of state-owned enterprises, particularly the power utility Eskom, which led to country-wide power shortages and rolling blackouts, and
- 2020 arose because of the COVID-19 pandemic and its associated declines on exports, foreign investment, and demand for South African commodities. Lockdown measures impacted small and medium-sized enterprises, forcing closures, widespread job losses, and a decline in consumer spending.

Charge-off rates may be converted to PDs by dividing them by relevant LGDs. LGDs are highly variable across loan types, individual obligors, and over time; thus, the assignment of a single-point LGD is non-trivial. Although LGDs used in this article were provided by the sources listed in Table 2, the literature is understandably silent (for proprietary reasons) on advanced approach LGDs for various loan types. To compensate for this shortcoming, a wide range of LGDs was used, i.e.,

$$80\% \cdot LGD \leq LGD \leq \max(120\% \cdot LGD, 100\%) \tag{13}$$

**Table 2.** Exposure types, LGDs, and sources used for the two regions.

| Region | Exposure Type | LGD (%) | Source |
|:---:|:---:|:---:|:---:|
| US | Residential mortgage | 30 | Neumann (2018) |
| US | Qualifying revolving retail | 40 | Ross and Shibut (2015) |
| US | Other retail | 85 | Banerjee and Canals-Cerdá (2012) |
| US | Corporates, sovereigns, and banks | 75 | Bandyopadhyay and Singh (2016) |
| US | High-volatility commercial real estate | 60 | Emery et al. (2009) |
| SA | Other retail, sovereign | 50 | Aslam (2020) |

Output variations are shown as error bars in Figure 4.

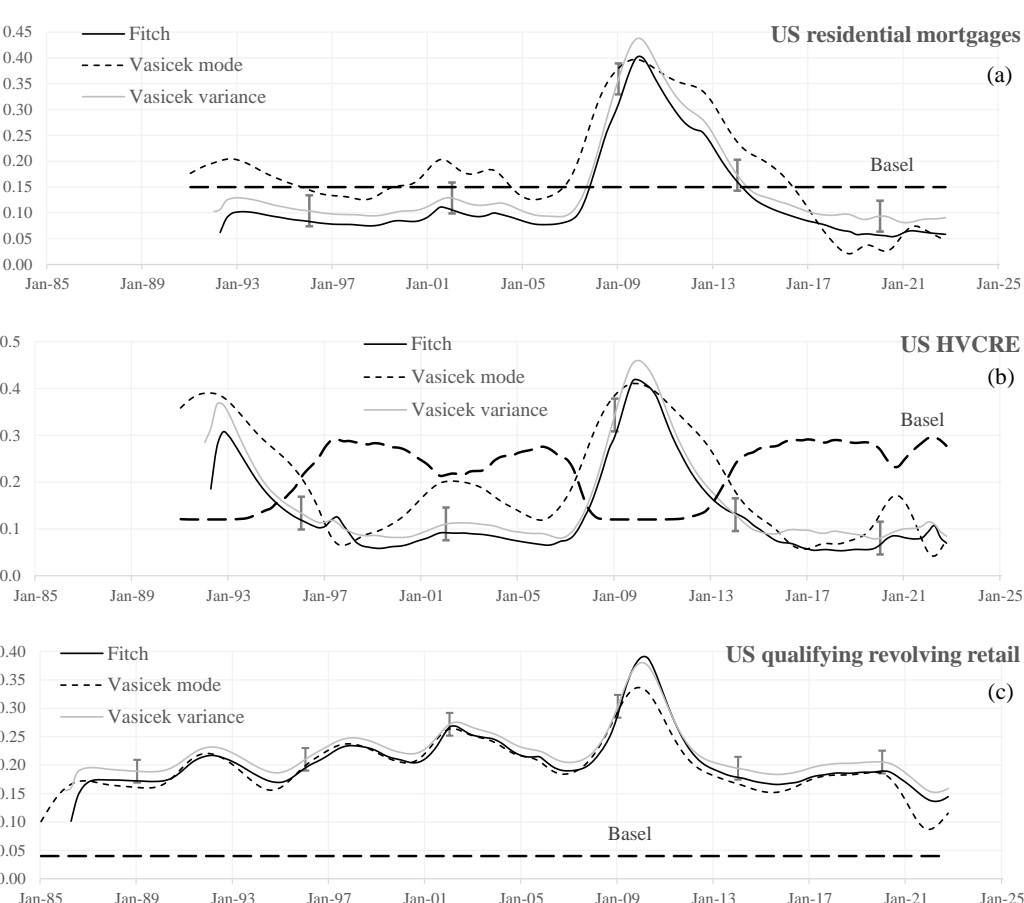

**Figure 4.** *Cont.*

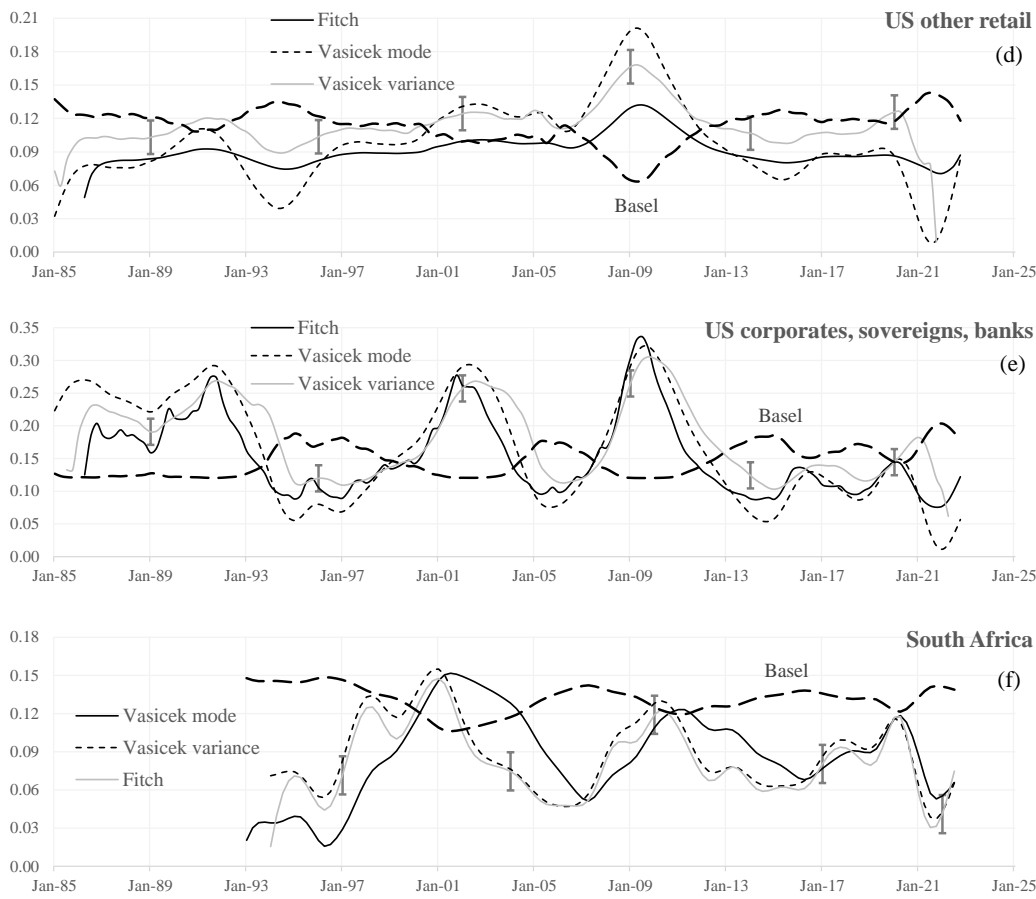

**Figure 4.** Implied asset $\rho$s derived from the $\beta$-fitting approach (Hansen et al. 2008) and the Vasicek portfolio loan loss distribution's mode and variance. (**a–e**) US loan types and (**f**) South African loans over a period which spans the 2008 credit crisis and the 2020/2021 COVID-pandemic. Error bars indicate the range of $\rho$s obtained for the range of LGDs set out in (13) from those values stated in Table 2. Sources: Hansen et al. (2008), Vasicek (2002), BCBS (2005) and authors' calculations.

Sample sizes of 12 quarters (three years) were used to measure the various requisite parameters ($L_{mode}$ (6) for calculating $\rho$ from (8), $s^2$ to reverse engineer $\rho$ from (9), and $\alpha$ and $\beta$ to determine $\rho$ from (12)). These samples were then rolled forward quarter by quarter to generate the relevant time series data of evolving $\rho$ values, shown in Figure 4a–f.

The asset $\rho$ magnitudes obtained for loan types which vary with PD, i.e., $\rho = \rho(PD)$ are similar (within established uncertainties indicated by error bars), while those obtained using the Basel ASRF approach are inverted.[1] Asset $\rho$ peaks for the three alternative methods are troughs under the Basel ASRF approach and vice versa; thus, the Basel ASRF methodology indicates lower asset $\rho$s during market turbulence (characterized by high default rates) and higher asset $\rho$s during calm periods (characterized by low default rates). All alternative approaches indicate the opposite. Basel ASRF asset $\rho$ profiles over time are, thus, both illogical and confusing, while those generated by the alternative approaches are sensible and consistent with the reasoning provided in Section 2 (after Figure 2). To reiterate, these approaches assert that asset $\rho$s increase with increasing PD over time, while the current regulatory formulation posits that the opposite occurs.

An MLE approach was used to recalibrate the Basel II (BCBS 2005) $\rho$ parameters such that they matched other approach results. Rather than duplicate many calculations, the $\beta$ distribution fitting approach (Hansen et al. 2008) was selected for the target asset $\rho$. The results are shown in Table 3 and the resulting Basel asset $\rho$s using the empirical parameters shown in Figure 5a–d for loan types whose $\rho = \rho(PD)$.

**Table 3.** Exponents and upper and lower bounds for $\rho$-fitted functions currently stipulated by Basel (Table 1) and derived empirically from loan loss data using an MLE approach. The loan types omitted (US residential mortgages and credit cards) are assumed to be constant, i.e., $\rho \neq \rho(PD)$. Source: BCBS (2005) and authors' calculations.

| Loan Type | Asset $\rho$s | Lower | Upper | Exponent |
|---|---|---|---|---|
| US HVCRE | Current (Basel) | 0.12 | 0.30 | 50 |
| | Recalibrated | 0.31 | 0.06 | 17 |
| US other retail | Current (Basel) | 0.03 | 0.16 | 35 |
| | Recalibrated | 0.20 | 0.06 | 20 |
| US corporates, sovereigns, banks | Current (Basel) | 0.12 | 0.24 | 50 |
| | Recalibrated | 0.29 | 0.05 | 13 |
| SA loans (other retail) | Current (Basel) | 0.03 | 0.16 | 35 |
| | Recalibrated | 0.34 | 0.02 | 33 |

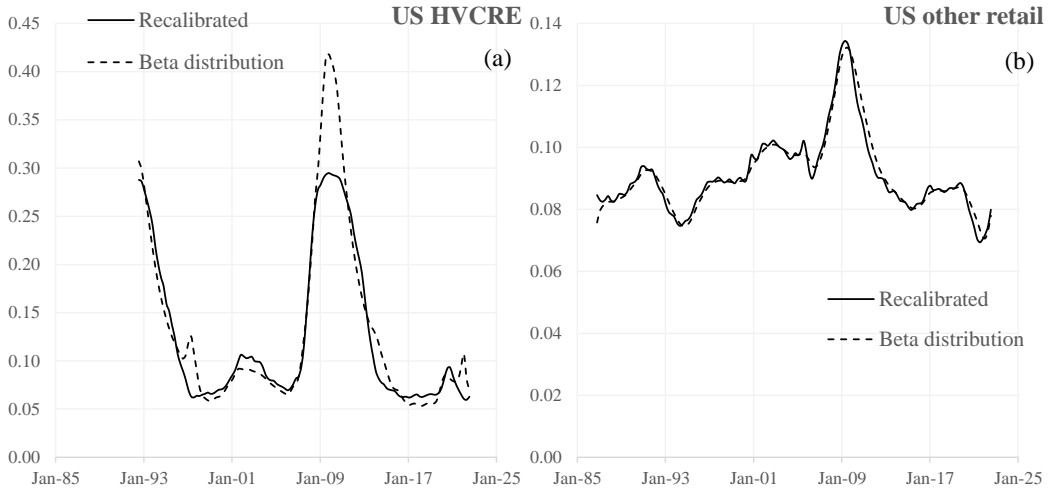

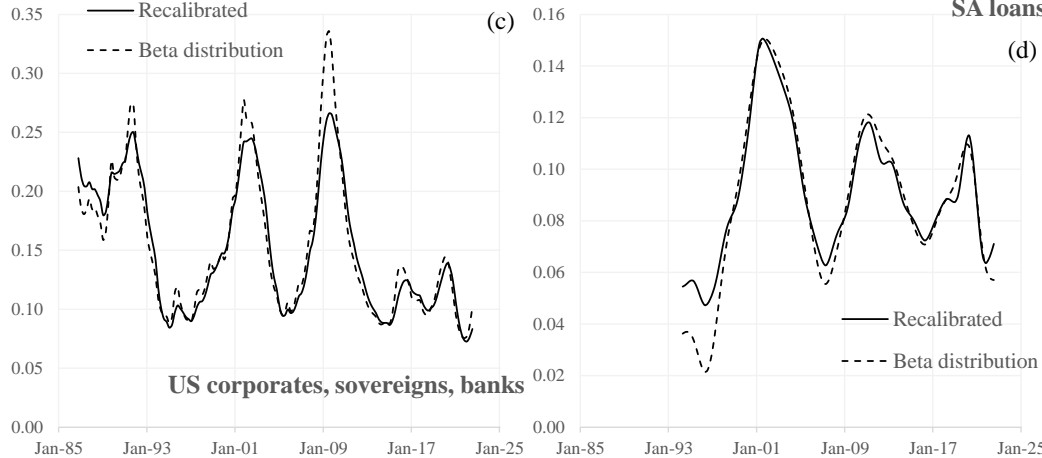

**Figure 5.** Recalibrated Basel $\rho$s after applying an MLS approach to differences between original parameters and those obtained using the $\beta$ distribution (Hansen et al. 2008). $\rho$s were arbitrarily selected: all three approaches generate similar empirical results.[2] (**a–c**) US loan-type results and (**d**) South African loan results. Sources: BCBS (2005) and authors' calculations.

Implementing the empirical asset $\rho$s will impact credit risk regulatory capital. A comparison of credit risk capital as determined by current regulatory rules and empirically determined $\rho$s are shown in Figure 6a–f for a hypothetical total loan portfolio exposure of

100, LGDs sourced from Table 2, maturities of one year (where applicable), and prevailing PDs derived from charge-off rates.

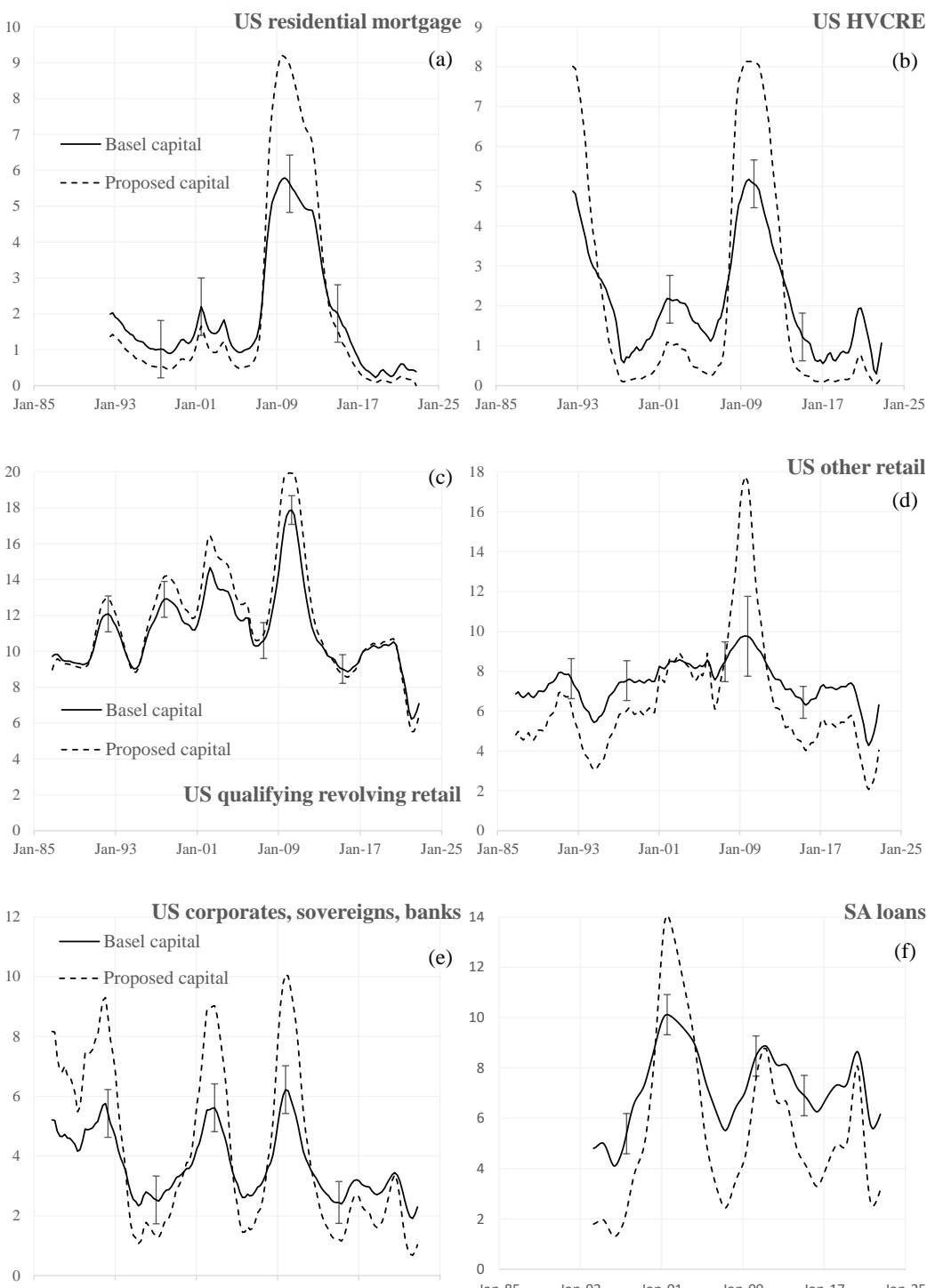

**Figure 6.** Comparison of regulatory Basel credit risk capital and empirically derived credit risk capital using for US loan types (**a**–**e**) and South African loans (**f**) over a period spanning both the 2008 credit crisis and the 2020/2021 COVID-pandemic. Error bars generated using the variability in asset $\rho$s derived from the range of potential $\rho$ values obtained for Figure 4. Sources: BCBS (2005) and author calculations.

The results indicate that capital levels are currently too low during market crises (periods of high default) and too high during calm market periods (low defaults). This is

an important result for risk mitigation as these capital levels are used to provide safety during market turbulence.

Lastly, a comparison of asset $\rho$ as a function of PD using current Basel requirements and empirically derived $\rho$s is shown in Figure 7a–f.

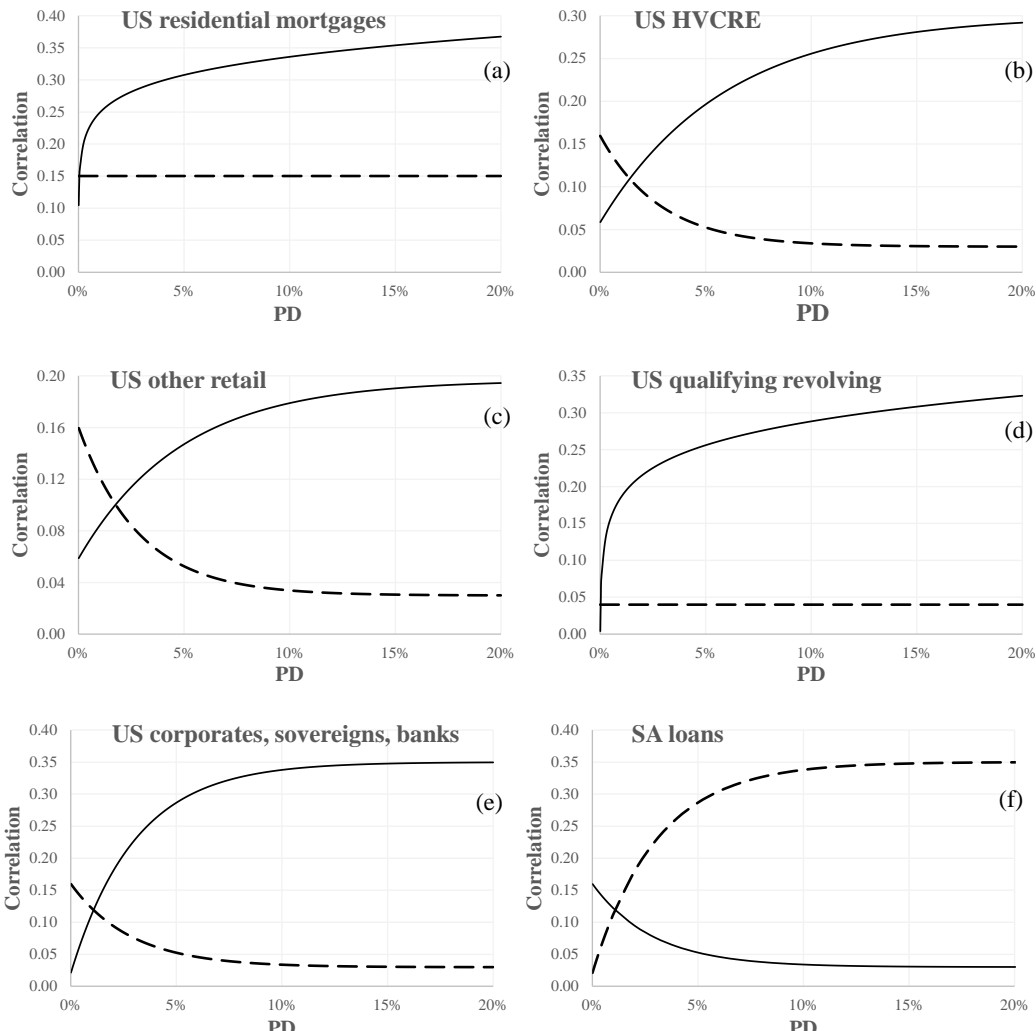

**Figure 7.** Regulatory Basel asset $\rho$s and proposed empirically derived values as a function of PD for (**a–e**) US loan types and (**f**) South African loans. Dashed lines indicate BCBS (2005) asset $\rho$s per loan type, while solid lines indicate empirically derived asset $\rho$s. Sources: BCBS (2005) and author calculations.

## 5. Conclusions

Compared with the three alternative methodologies to estimate asset $\rho$s, prescribed BCBS asset $\rho$s are countercyclical to empirical loan loss experience over the entire 30-year period of loan loss experience investigated. The original assessment, calibration, and assignment of asset $\rho$s—set before the introduction of Basel II in 2008—are flawed and result in less punitive credit risk capital requirements than required during and after market crises and more punitive than required in calm conditions. The original calibrations, therefore, are redundant and should be updated and incorporated into the Basel framework. It is possible that the misalignment of Basel's estimated asset $\rho$s may understate default levels during periods of market turbulence, surely an unintentional regulatory consequence. The approaches detailed in this article do not guarantee consistency as the market will eventually reveal limitations embedded in current models. This work, however, aimed

to demonstrate inconsistencies in the *current* asset correlation formulation, prescribed by Basel for regulatory capital allocation.

### 5.1. Limitations

In the absence of default data distinguished by loan type, losses arising from charge-offs are assumed to be a suitable and reliable proxy for default rates. Charge-off data for only one developed (US) and one developing (South Africa) country were used in this analysis. A principal reason for this is the scarcity of granular default rate data. Time series of default rates are available, but these are recorded at different frequencies (some only annually) for different jurisdictions, and—even where available—losses distinguished by loan type are absent.

Loan losses are assumed to be distributed either according to a Vasicek's (2002) formulation or a $\beta$ distribution. Both come with assumptions and limitations, thus influencing the results.

LGDs are assumed to be relatively constant, and, in the absence of more regular research governing the evolution of this important parameter, existing results were used, some published over a decade ago. Although this limitation was partially ameliorated by generating outputs from a range of sensible LGDs, current LGDs relevant for each jurisdiction should ideally be used.

### 5.2. Future Work

Future work could extend this analysis to a wider base and include more—and different—country-specific loan loss data. Where available, loan-type data (measured as frequently as possible) should be employed, and, if absent from a national database, their assembly and recording should be encouraged.

Other distributional assumptions could be used, and relevant percentile losses could be extracted from these for comparison with empirical loss experience. Historical credit risk capital requirements, determined using the BCBS equations, could be compared with proposed values and assessed against historical loss experience to ascertain and evaluate capital adequacy for credit risk losses.

Because higher asset correlations result in higher unexpected losses and associated higher capital charges, if all banks in a jurisdiction are undercapitalized due to inaccurate asset correlation values, systemic risk could be the result. Although Basel III requires changes to both the quality and the quantity of regulatory capital, this takes the form of capital buffers and increased tier 1 capital requirements rather than changes to the credit risk capital formula or adjusted asset correlations. We recommend that banks raise the level of their economic capital to adjust for the regulatory shortcomings of the current Basel-mandated approach.

**Author Contributions:** Conceptualization, A.v.D. and G.v.V.; methodology, A.v.D. and G.v.V.; software, A.v.D. and G.v.V.; validation, G.v.V.; formal analysis, A.v.D. and G.v.V.; investigation, A.v.D. and G.v.V.; resources, G.v.V.; writing—original draft preparation, A.v.D. and G.v.V.; writing—review and editing, G.v.V.; visualization, G.v.V.; supervision, G.v.V.; project administration, G.v.V. All authors have read and agreed to the published version of the manuscript.

**Funding:** This research received no external funding.

**Data Availability Statement:** Data used in the study available from https://fred.stlouisfed.org/ (accessed on 12 April 2023).

**Acknowledgments:** The authors are grateful for fruitful discussions with colleagues in the Mathematics Department and at RiskWorx.

**Conflicts of Interest:** The authors declare no conflict of interest.

## Notes

[1]   Positive solutions of (8) give asset $\rho$s whose time profile is consistent with those specified by the BCBS (2005), but *considerably* higher.

[2]   Indeed, new Basel parameter values obtained using other methodologies differed only from those obtained in Table 3 by a few percent.

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
