# Peer review of "Measurement and Calibration of Regulatory Credit Risk Asset Correlations"

_jrfm, doi:10.3390/jrfm16090402_

Round 1

Reviewer 1 Report

See the attached referee report. Mainly it should be clear and focusing on the $\rho$ definition, calibration and measurement.

Author Response

We are grateful for the helpful and insightful comments.

Please see attached document and amended manuscript for changes made.

Reviewer 2 Report

The research is focused on a current issue that affects the stability of banks and the banking system. In order to improve the proposed paper some corrections could be made.

First of all, the introduction could be shortened as the analytical part of it including formulas and figures should find a place later in the exposition of the paper.

At the end of the introduction, for clarity and better orientation of the readers, a brief description of the logical sequence of the content in the following sections of the paper should be presented.

It is noteworthy that the main emphasis of the paper is on the Basel II capital accord. The impact of the new regulatory requirements in Basel III, which are not mentioned in the paper, should also be considered. For example, to analyze how they would change the graphs in Figure 6. 

The authors convincingly justify their research's conclusion that "capital levels are currently too low during market crises and too high during calm market periods." We do not comment here on the limitations of the study regarding the database used and their relevance. However, it is considered appropriate after presenting the research results to formulate specific proposals, recommendations or guidelines to overcome the identified problem.

Author Response

(The authors gave the same response as above.)

Round 2

Reviewer 1 Report

1. Line 139, Zhou's method to understand the default correlation is improper with tremendous calculation errors. Hence Li and Krehbiel (2016) indicate Zhou's method to understand the default correlation is inconsistent, not the stochastic assumption of Merton's firm-specific model inconsistent.

2. Line 330, the rho for the asset correlation between one underlying asset and the systematic risk factor from an economic index, what are the possible borrowing company's specific risk factor? Is it diffusion factor of the company's asset or the default risk? Calculations in Table 3 go further to show that the $\rho$ is actually a function of probability of default. Formula (12) on line 398 presents a different function. There are lack of logical explanations about these different rho's! Hard to get the point without understanding the rho's!

3. Line 466, the sentence requires further explanations on the rho function over time, not over the probability of default; the rho function illustrates some illogical and confusing issues, what are these issues? It would be better to clearly identify these for people who use Basel ASRF in daily basis. Solving the rho from the inverting process has the same issue of consistency, how do you guarantee this? 

Reviewer 2 Report

Thanks to the authors for the detailed responses to the recommendations. I believe that the revisions made improve the quality of the paper.

In terms of Basel 3, really the latest version of the Capital Accord does not offer anything new in terms of asset correlation and I think that should be mentioned in the paper. However, I remain of the opinion that Basel 3 is relevant to the problem because of the higher capital requirements imposed.
